# The Influence of Commodity Presentation Mode on Online Shopping Decision Preference Induced by the Serial Position Effect

**Zhiman Zhu, Ningyue Peng** 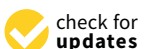**, Yafeng Niu, Haiyan Wang and Chengqi Xue** *

School of Mechanical Engineering, Southeast University, 79 Suyuan Avenue, Nanjing 211189, China; 220190378@seu.edu.cn (Z.Z.); pengny@seu.edu.cn (N.P.); nyf@seu.edu.cn (Y.N.); why-box@163.com (H.W.)
* Correspondence: ipd_xcq@seu.edu.cn; Tel.:+86-025-52090510

**Abstract:** The information cluster that supports the final decision in a decision task is usually presented as a series of information. According to the serial position effect, the decision result is easily affected by the presentation order of the information. In this study, we seek to investigate how the presentation mode of commodities and the informativeness on a shopping website will influence online shopping decisions. To this end, we constructed two experiments via a virtual online shopping environment. The first experiment suggests that the serial position effect can induce human computer interaction decision-making bias, and user decision-making results in separate evaluation mode are more prone to the recency effect, whereas user decision-making results in joint evaluation mode are more prone to the primacy effect. The second experiment confirms the influence of explicit and implicit details of information on the decision bias of the human computer interaction caused by the serial position effect. The results of the research will be better applied to the design and development of shopping websites or further applied to the interactive design of complex information systems to alleviate user decision-making biases and induce users to make more rational decisions.

**Keywords:** serial position effect; information presentation; evaluation mode; decision-making biases; consumer decision-making; decision support systems

## 1. Introduction

With the rapid development of computer technology and information technology, online shopping has long been integrated into people's daily life, and it is also one of the high-frequency decision-making tasks that cannot be avoided in life. When a consumer searches for a product and finally makes a purchase decision, they need to browse 124 different product pages on average [1]. During any product browsing process, the user is bound to receive, compare and judge a large amount of product information. Information on each product also contains different product attributes and evaluations. Due to limited cognitive ability, the human brain's ability to recognize, select and filter decision-making information; pay attention to, perceive and benefit from information processing and remember it is quite limited [2]. When users encounter products that are encountered continuously, their cognitive ability will gradually be consumed over time, and it is not easy to maintain the same treatment of information presented in different sequence positions. This phenomenon is called the serial position effect in the field of psychology. Erik Maier (2019) put forward three factors that affect the position effect of sequences in product evaluation, establishing the position of sequences as a structural driving factor for the sequence evaluation of different products in a category [3].

In different environments, the serial position effect also changes with the changes of response mode, decision interval, stimulus duration, sequence length, information density, audio–visual channel, etc. [4–6]. In the past, researchers have tried to reduce decision-making bias by extending feedback time, shortening the length of information,

increasing auditory channels, etc. Although these methods have been proven effective in some cases, they are not suitable for all environments. Compared with offline shopping, online shopping is more affected by fixed interface interactions, with a fixed presentation method and a fixed length of information, and the duration of user interaction feedback is difficult to control manually. Therefore, under the application conditions of the human–computer interaction interface, the research on the factors that can still affect the change of the serial position effect is currently lacking. This article chooses a more versatile factor—*views* (the information presentation method)—as the entry point for the factors that cause human perceptual salience to be highlighted by information visualization [7]. Furthermore, the reversal phenomenon of the human–computer interaction on the decision-making preference induced by the serial position effect and caused by the change of information presentation mode is studied.

In this study, we combine the serial position effect with ergonomics, not only focusing on the decision style of a certain type of consumer or a certain category of products but also on the influence of the information presentation method on the serial position effect in a broad sense.

Through the study, we seek to answer the following research questions:

RQ1: Will the serial position effect induce user decision-making bias?

RQ2: Will the change of views in information visualization affect the decision-making bias in human–computer interaction induced by the serial position effect?

RQ3: What role do the factors that make up the different views in information visualization play in the impact?

To give answers to the three questions, we conducted two experiments. Experiment one is to verify RQ1 and RQ2. The comprehensive analysis of Experiment one and Experiment two is to verify RQ3.

## 2. Literature Review

### 2.1. Serial Position Effect from the Online Shopping User Interfaces

The serial position effect can broadly be defined as follows: when a person recalls a sequence of things, the memory at the beginning and the end of the sequence is far more vivid than the information in the middle [8], which includes the primacy effect, recency effect and peak effect. It was proven that items in the beginning and the end of a list were recalled better and more frequently than those in the middle of the list [9]. It was a concept that was first raised against the psychology domain by using the free recall paradigm, but it was confirmed in domains of sociology, driving attribution, user experience, group decision-making, anti-air warfare (AAW), etc. [10–14].

The influence of the serial position effect on a user's decision making is usually shown by the ignorance of the visual content in the middle part of the free recall, which forms a false impression. As a result, the retrospective evaluation (RE) of the previous information forms an error, which affects the user's understanding and evaluation of the data trend represented by the visual presentation sequence. Ayton P (2015) suggested that memory-based analysis can predict the response of event sequences; that is, the use of a memory-based perspective helps researchers to understand the determinants of RE in time series more fully [15]. As many scholars have discovered, certain moments or characteristics of an event are more important than other moments or characteristics in RE [16]. The main explanations for these findings include memory-based methods, which show that part of the bias in decision-making bias can be attributed to the availability of biased moments in memory [17]. With the passage of time, the amount of information increases. When people rely on information before making a decision, the information cannot maintain the equality of when it was presented, and people must rely on memory retrieval [18]. Many studies indicate that in different application scenarios, the effect of the serial position effect will also vary with changes in factors such as the response mode, decision interval time, stimulus time, sequence length, information density and audio–visual channel changes [4–6].

These selection decision biases, affected by the serial position effect, are found to be applicable to many different types of consumer experience [19]. In consumer psychology-related research, researchers mostly use a specific task scenario as the basis [18,20,21], combined with the psychological experimental paradigm and consumer application context, to study the influence of the serial position effect on user decision-making bias in this task context. Most of these studies were limited to the comparison of traditional shopping tasks and rarely studied the inducing effect of different forms of information visualization from the perspective of ergonomics. However, the User Interface (UI) design of a website in online shopping is a key part that affects the user's decision-making; this is still lacking in the existing research, and it is also the research focus of our study.

### 2.2. Factors Influencing Online Shopping Decisions

At present, research on decision-making bias in information visualization combined with the concept of psychology mainly focuses on the more common psychological effects, such as the attraction effect, anchoring effect and primer effect [22–25]. In information visualizations, the perceptual salience of the form of information representation on a computer to human cognition is summarized into six levels from the global to the local [7]. Forms and views are both basic influencing factors for the influence of information visualization on user decision making. The difference between the two is that "form" refers to how key attributes and data are mapped to visual elements and what kind of graphical form is described or designed, but views are a more macro-level concept, which is more widely used in the presentation of the generated form. Since form is itself more restricted by data and tasks, it needs to be studied for more specific data, and the research is not as basic as the views. In an online shopping environment, it has long been confirmed by research that the presentation mode of a product, can induce decision-making bias by influencing consumer emotions [26]. Therefore, this research will focus on views first, and use online shopping scenarios as the experimental environment to conduct research.

The Internet is a tool for dialogue between suppliers and consumers, and its ability to convey key information affects the quality of the relationship between suppliers and consumers [27]. Especially since the COVID-19 pandemic, online shopping has become the mainstream form of consumption [28]. Limited by the operation of electronic devices and shopping websites, text information and image presentation have become the main types of information in online product displays [29]. Properly combining text and images in a coherent page layout is a recognized and critical aspect of complex information presentation [30]. Referring to the data display grouping content in components in Ant Design Vue and existing shopping website applications, we summarized and selected views in the shopping environment as follows: (a) table; (b) collapse; (c) card; (d) carousel; (e) tree; (f) popovers (see Figure 1 for details).

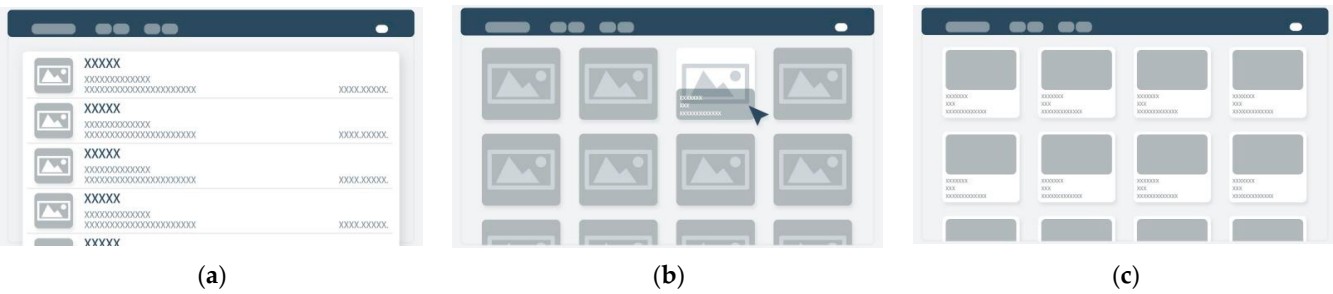

(**a**)                    (**b**)                    (**c**)

**Figure 1.** *Cont.*

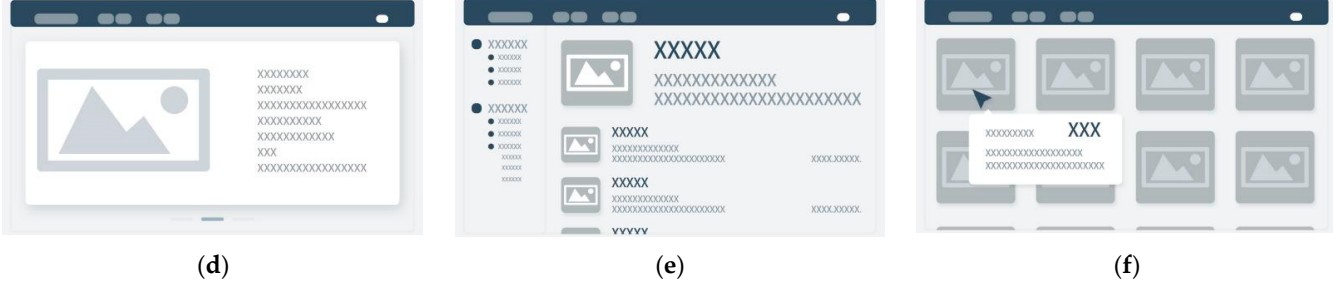

**Figure 1.** Six commonly used view types in online shopping websites. Subfigure: (**a**) table; (**b**) collapse; (**c**) card; (**d**) carousel; (**e**) tree; (**f**) popovers.

These six views are common in existing shopping website UI applications. Card, collapse and carousel are more basic, are not restricted by category and context and are usually used to display product search results in a large area. Moreover, the kind of impact these basic presentations have on the user's cognition and memory process and whether they constitute a change in the user's decision-making results is the research focus of this work.

### 2.3. Evaluation Mode

According to general evaluability theory, all judgments and decisions in the shopping environment are made in one (or some combination) of the two basic evaluation modes. Joint evaluation mode (JE) refers to multiple options appearing at the same time and the use of comparative evaluation; separate evaluation mode (SE) refers to multiple options appearing separately that are evaluated separately [31]. Compared with SE, JE has more usability comparisons between alternatives in the presentation of information and greater measurement evaluability; compared with JE, since SE has less contextual information, its additional information will gain greater decision weight [32].

SE and JE are two continuums that change according to different situations. General Evaluability Theory, proposed by Hsee (1996), explains that preference reversal occurs between JE and SE due to the differences in the evaluation mode itself [31]. There are also results indicating that, under SE, consumers may be relatively dependent on feelings and emotions when making decisions, while people under JE may rely on theory and logic [33]. Researchers in many fields have also confirmed the obvious reversal of decision preferences under different evaluation modes. These studies involve the context effect, frame effects, Weaver, Garcia and Schwarz's presenter's paradox, multi-attribute evaluation, etc. [34,35].

It is obvious that scholars usually prefer to involve the two extremes of this continuum in experimental research; that is, the most "separate" evaluation and the most "joint" evaluation in the shopping environment form the maximization of the contrast between the experimental results. Continuing the concept of the view summarized in Section 2.2, the card (waterfall flow layout) view belongs to the extreme JE, where multiple options appear continuously simultaneously, while carousel is an extreme SE in which multiple options appear independently. These two views perfectly correspond to the two extreme values of the evaluation mode. This study proposes that the following two factors constitute the gap between these two evaluations on views: the method of information presentation and quantity of information. By controlling the levels of these two factors, we can adjust the state of the evaluation mode. Information presentation means the extent to which the context in the decision-making environment affects users; when context effects occur, individuals tend to make relative decisions that are affected by the environment or previous contact with objects [36,37]. Quantity of information means the amount of information presented at a single time; it also refers to the amount of information conveyed through an online product display. The information load caused by this affects information processing due to people's limited cognitive ability [29]. It is an essential factor that adjusts the intermediate value of the two extreme evaluation modes.

*2.4. Research Hypotheses*

According to the previous theoretical analysis, the influence of views on the serial position effect inducing the user's decision-making bias mainly occurs in the memory stage. The memory formed by a user in SE is a series of independent impressions; it is minimally affected by the context and transforms into long-term memory with the continuous refreshing and accumulation of subsequent information [38]. In this environment, when the user reconstructs a memory reading before the final decision, as the end target stimulus remains in the working memory, it is more likely to have a recency effect than the intermediate stimulus [39–41]. In JE, the memory formed by the user is comparative. The user does not refresh the interface view in the continuously presented information group. Therefore, the user's memory of the target stimulus is relatively integrated. Compared with a stimulus that appears later in the same sequence, the stimulus presented in the first paragraph has undergone a more extensive rehearsal, which leads to better consolidation and long-term tracking comparison. It is more likely that there will be a primacy effect [42,43]. Of the factors that constitute the evaluation mode, information presentation is the most important factor. In theory, the quantity of information and collapse will also change the degree of influence of the context. They can each be used as a variable for experimental comparison to observe their effects and be used as a noise factor to eliminate the influence on information presentation. Based on the aforementioned review, we propose three hypotheses that corresponding to the research questions, respectively (The experimental relationship corresponding to the research question is shown in Figure 2).

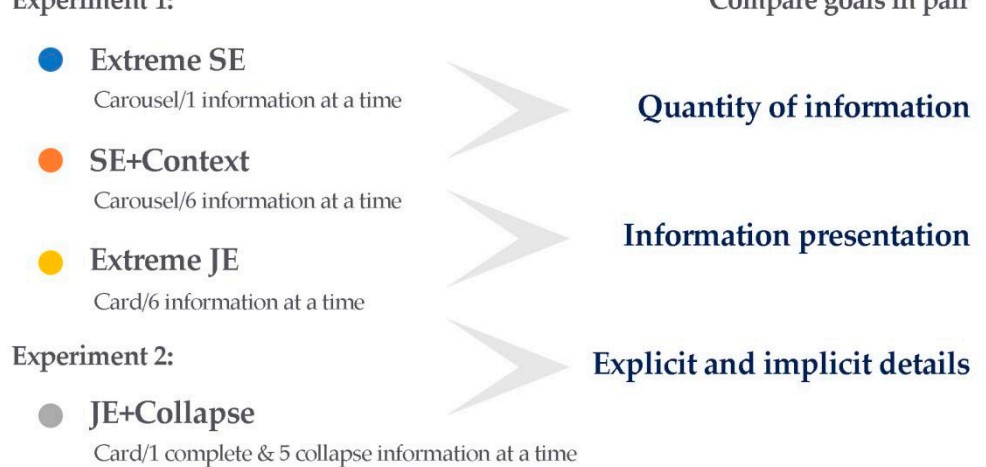

**Figure 2.** The relationship between the experimental groups in Experiment 1 and Experiment 2.

**Hypothesis 1 (H1).** *The serial position effect in the online shopping environment will induce user decision-making bias, and the user's evaluation mode and their preference will be reversed as the view type changes.*

**Hypothesis 2 (H2).** *The change of views in information visualization affect the decision-making bias in human–computer interaction induced by the serial position effect. User decision-making results under SE will be more prone to the recency effect, while user decision-making results under the JE mode are more prone to the primacy effect.*

**Hypothesis 3 (H3).** *The three factors that constituted different evaluation modes: information presentation, the quantity of information, and folding of details, play important roles in preference reversal.*

## 3. Research Methodology

### 3.1. Apparatus and Participants

#### 3.1.1. Apparatus

The experiment used the questionnaire star web version to collect decision data and conduct a subjective evaluation. The Tobii X2-30 eye tracker, with a sampling rate of 30 Hz, was used to collect eye movement data. The resolution of experimental picture material was 1920 px × 1080 px, while that of web experimental material was 1920 px × 2760 px. The picture experiment was programmed with Tobii Studio 3.2.3. and the web experiment materials were compiled in Adobe XD 34.0.12.33. All experimental materials were displayed on an HP 24-inch display. The laboratory was set up under normal lighting conditions (40 W fluorescent). The viewing distance ranged approximately from 600 to 700 mm.

#### 3.1.2. Participants

Thirty participants were enrolled (17 female and 13 male participants, ranging from 20 to 28 years of age). Among them, one person lost data due to system problems, and one person was ruled out due to poor performance due to data confusion. (In this experiment, the three optimal values in the objective sense were not selected as irrational decisions. If the percentage of incorrect data exceeded 10%, this was considered to represent unsatisfactory performance; this participant scored 42.3%.) There were 28 groups of valid data in the experiment. The experiment was conducted at Human Factors and Ergonomics Laboratory of Southeast University. Participants were compensated CNY 40 for their participation.

### 3.2. Experiment Environment

#### 3.2.1. Simulated Web Purchase

The actual shopping environment will be affected by many factors. These influences come from the environment, personal preferences, product category characteristics and other aspects. Each factor is sufficient to affect the final selection results of the subjects. Therefore, we needed to eliminate the distracting factors and design and construct a virtual shopping scene for our research purposes. In the virtual scene, we designed each individual stimulus to contain the following four pieces of information: product picture, product serial number, product attributes and product price. The product picture and serial number did not affect the user's decision-making results in this experimental setting. The function of the product picture was to build a shopping environment (the shopping environment without pictures was too abstract, and it would be difficult for the subjects to enter the task with a sense of substitution), while the product serial number served to provide participants with memory and feedback and to simulate the process of additional purchases during shopping. In order to reduce the impact of images on user decision-making, we processed the displayed product images and replaced the authentic product images with vector icons of the same color drawn using Adobe AI. In addition, before the experiment, the following was emphasized to subjects: "In this experiment, the subject's personal preference for pictures cannot affect the final decision-making results and is only used to build a shopping environment".

In the virtual environment of this experiment, the only two factors that affected the user's decision-making were attributes and prices. The basic task of the experiment was to comprehensively evaluate the entire sequence to select the most high-quality and cheap products (the lower the price is, the better, and the higher the property is, the better). Considering the nonalignable differences in structure-mapping theory [44,45], we quantified the factor levels of the two factors [24]: the attribute value was set to four levels, and the price value was set to five levels. In order to fit the actual situation as much as possible, we converted the information of a single product into three common product attributes with a total score of five. Since multi-attribute decision making was not the focus of this experimental study, we spoke with the subjects in the experiment and asked for the three attributes to be treated with equal weight; that is, the attribute value should be

regarded as a whole with a full score of 15. The product price in the environment was selected from the price range according to the actual pricing of the product on the shopping website, and we divided the prices in the range into five levels. Moreover, the product price included single digits to six digits, integers and decimals. We attempted to restore the actual online shopping scene based on the level of artificial control factors.

### 3.2.2. Optimal Value Setting

In this research, we aimed to ensure that the results observed in the experiment were only affected by the change of position; therefore, we needed to ensure that the attributes of the target options were equal except for the position. Therefore, in the experimental design stage, we needed to set three optimal values (corresponding to the primacy, middle and recency effects in the concept of the serial position effect) with the same attributes and arrange them in different positions for the subjects to choose from (if the user's attention to the three locations were equal, the probability of selecting each location should be 33.3%). Since the weight between attribute and price involves personal preference, it was not easy to subjectively define the proportion of the two. Only when the price and attribute were to reach the optimal value in this group simultaneously would the most undisputed optimal value be obtained; therefore, our optimal value was set as the product with the highest attribute value based on the lowest price value in the whole group.

### 3.2.3. Deployment of Sequence Locations

The key to this experiment was setting the optimal value to include the possible scenarios more comprehensively. We first divided the two categories into the following two categories: the distance between the optimal values as equal (18 orders), and the distance between the optimal values as not equal (12 orders). Then, by adjusting the position of the intermediate value to ensure that the position of the optimal value was covered in the overall sequence, and on this basis, the distance between the optimal values was adjusted to determine the position of the first and last optimal values (Figure 3a). Finally, the 30 set sequences were randomly shuffled in Excel, and 30 sets of random sequence position deployments used in the final experiment were obtained (Figure 3b).

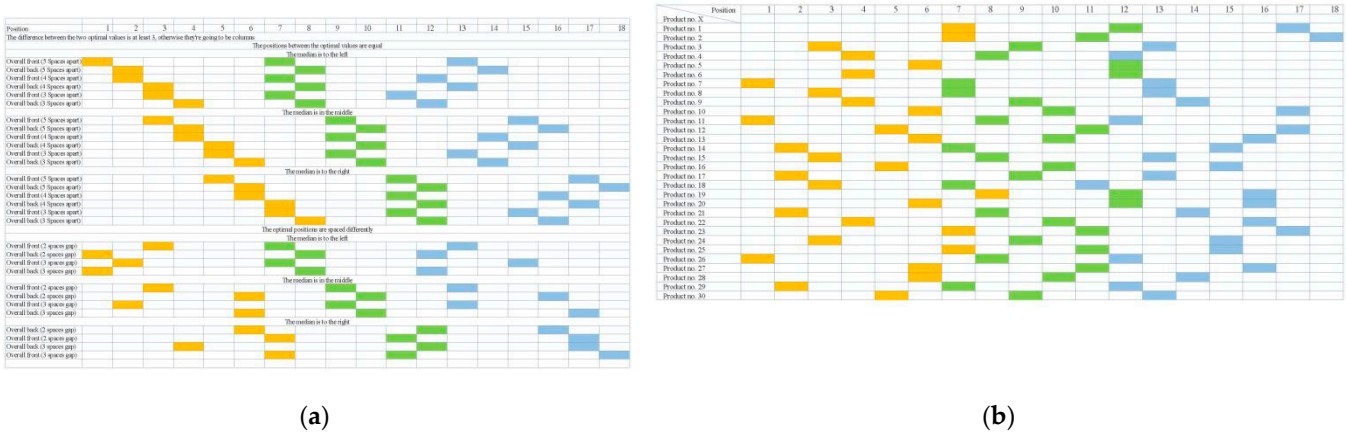

(**a**)　　　　　　　　　　　　　　　　　　　　　　　　　(**b**)

**Figure 3.** (**a**) is the setting sequence of the three optimal values in the experimental design, (**b**) is the actual application sequence after the setting sequence was randomly shuffled.

### 3.3. Stimuli Design

The experiment simulated a total of 30 product categories whose appearance had little impact on consumers. In each category, 18 products with different shapes, different prices and different attributes were produced. Then, we designed a simple website prototype map concerning the online shopping interface of the actual application and filled the above-mentioned experimental information into the prototype map to generate interactive website links.

### 3.4. Independent Variables and Dependent Variables

In addition to the presentation position of the information sequence (the presentation position of the optimal value in the sequence), Experiment 1 contained the following two independent variables: information presentation and quantity of information. Experiment 2 contained one independent variable, explicit and implicit details, and consisted of four evaluation modes. Thus, there were four experimental groups. In this experiment, a total of 30 non-repetitive stimulus groups were prepared, with each group containing 18 individual stimuli, and each group had four evaluation modes to present. Since a subject could not repeatedly make different evaluation mode decisions under the same stimulation group, we needed to recruit at least four groups of subjects for the experiment.

## 4. Experiment One

### 4.1. Objective

The primary objective of the experiment was to explore whether the human–computer interaction decision-making preferences induced by the serial position effect would be reversed under the evaluation mode transition. Moreover, we wanted to determine whether the two factors that constitute the evaluation mode and the view transition would show their respective roles in this reversal.

### 4.2. Experiment Setting

The 28 subjects were divided into four groups, and each group were subjected to three experimental tasks in different modes (the same stimulus could only be presented in one mode. For example, if the product "screw" was presented in the Extreme SE mode in Group A, it could not be presented in another mode in Group A). Mode one was Extreme SE, represented by the carousel view, and the number of pieces of information was one; Mode two was SE + Context, for which alternatives were added based on Mode one, and the number of pieces of information was six; Mode three was Extreme JE, represented by the card view (waterfall flow layout), and the number of pieces of information was six (the specific grouping is shown in Table 1, and the three different modes of the single product category are shown in Figure 4).

**Table 1.** Experiment 1, specific grouping table.

| Experimental Group (7 Participants Each) | | Simulated Serial Number | | | | | | | | Information Independence | Quantity of Information |
|---|---|---|---|---|---|---|---|---|---|---|---|
| Group A | Mode 1 | 1 | 2 | 3 | 4 | 5 | 6 | 7 | | Carousel | 1 |
| | Mode 2 | 8 | 9 | 10 | 11 | 12 | 13 | 14 | | Carousel | 6 |
| | Mode 3 | 23 | 24 | 25 | 26 | 27 | 28 | 29 | 30 | Card | 6 |
| Group B | Mode 1 | 8 | 9 | 10 | 11 | 12 | 13 | 14 | | Carousel | 1 |
| | Mode 2 | 1 | 2 | 3 | 4 | 5 | 6 | 7 | | Carousel | 6 |
| | Mode 3 | 15 | 16 | 17 | 18 | 19 | 20 | 21 | 22 | Card | 6 |
| Group C | Mode 1 | 15 | 16 | 17 | 18 | 19 | 20 | 21 | 22 | Carousel | 1 |
| | Mode 2 | 23 | 24 | 25 | 26 | 27 | 28 | 29 | 30 | Carousel | 6 |
| | Mode 3 | 8 | 9 | 10 | 11 | 12 | 13 | 14 | | Card | 6 |
| Group D | Mode 1 | 23 | 24 | 25 | 26 | 27 | 28 | 29 | 30 | Carousel | 1 |
| | Mode 2 | 15 | 16 | 17 | 18 | 19 | 20 | 21 | 22 | Carousel | 6 |
| | Mode 3 | 1 | 2 | 3 | 4 | 5 | 6 | 7 | | Card | 6 |

### 4.3. Experiment Procedure

Before the experiment, the principal examiner gave detailed explanations and training to each subject to ensure that each subject did not have any doubts about conducting the formal experiment. Participants in Mode one and Mode two were asked to complete a

Likert seven-point scale to assess their willingness to purchase each product. Moreover, after all 18 products of the same category were presented, the interface entered a blank page, and the participant needed to recall the serial number of one of the products with the best quality and the lowest cost (see Figure 5 for details). In Mode three, the subjects were asked to slide the mouse wheel to view the product information continuously ("try not to look back at the stimuli you have seen"). After browsing, they entered a blank page to recall and fill in the serial number of only the most cost-effective product for an additional purchase.

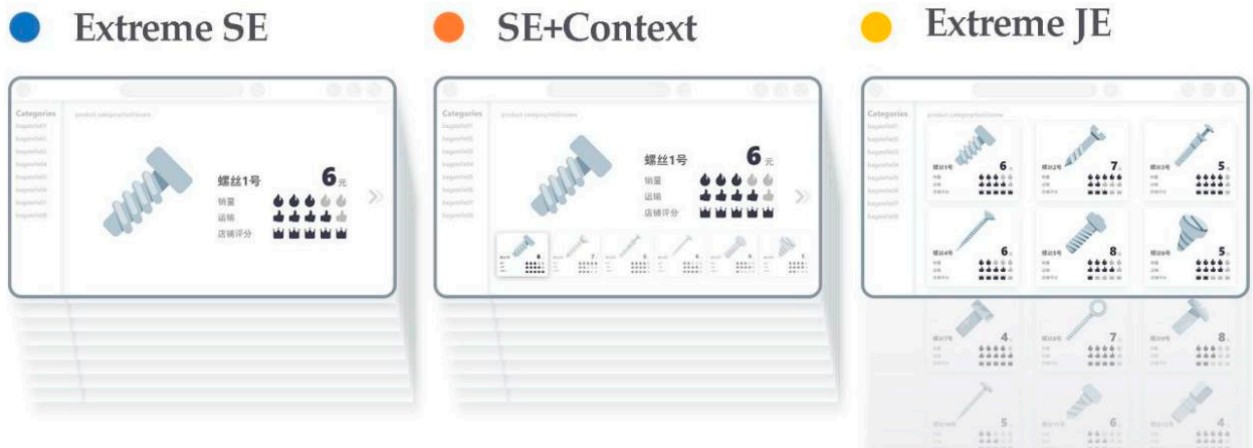

**Figure 4.** Three different modes of the single product category.

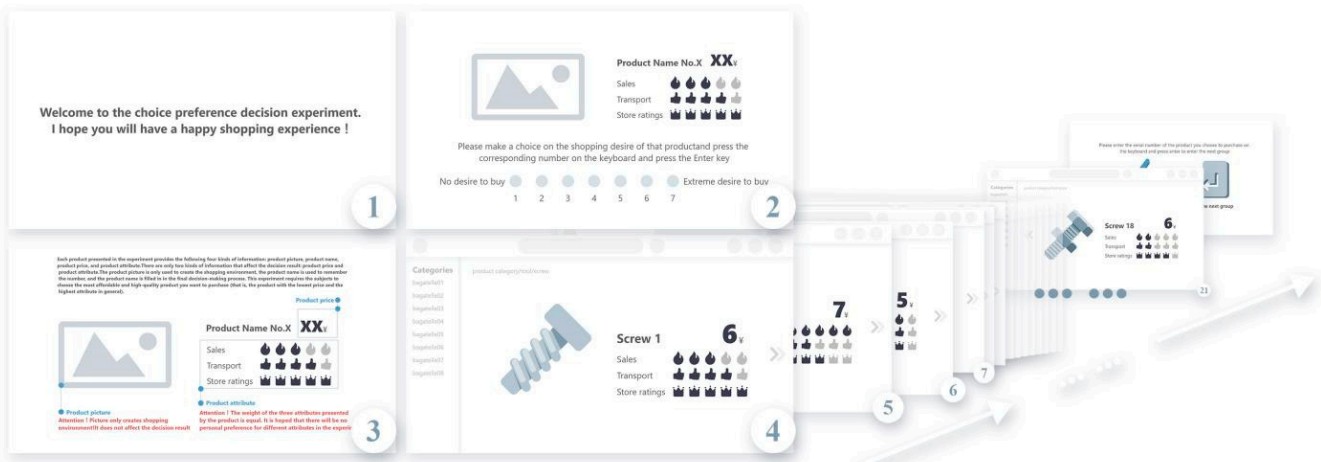

**Figure 5.** Schematic diagram of experimental task flow.

### 4.4. Analysis and Results

Excluding the 30 extremely irrational decisions (the three objectively optimal values of the stimulus set were not selected), we counted the probability distributions of the three optimal values of the choice preference of each participant in each mode in the remaining 516 pieces of decision data (Figure 6). Observing the distribution of the overall three modes horizontally, one can observe that the subjects' preferences for primacy or recency were much higher than the middle (if the user's attention to the three locations were equal, the probability of selecting each location should be 33.3%), which fits the "U"-shaped serial position curve of the serial position effect. As a result of the differences between the three groups of different modes, it can be observed that the recency effect of the first two groups was higher than the primacy effect, and the preference of the primacy gradually increased until Mode three realized the inverse, with the primacy effect higher than the recency effect.

Observing the three positions longitudinally, there are noticeable differences among the three effect groups for primacy, and the preference for primacy steadily increased from Mode one to Mode three; the difference between the middle group is not noticeable, but it can be observed that the user's preference for the middle steadily decreased from Mode one to Mode three; the difference between Mode one and Mode two is not apparent for recency, and Mode three is much lower than the previous two modes.

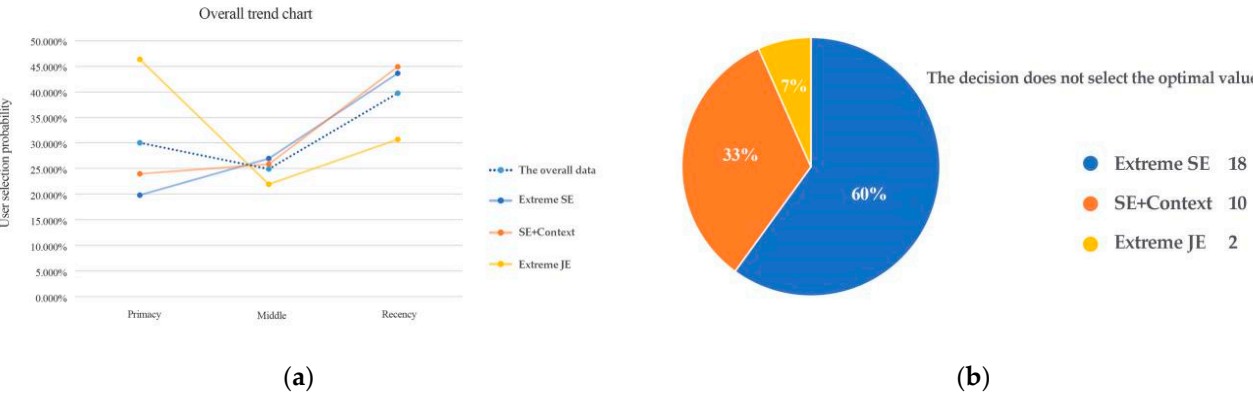

|     |     |
| :-: | :-: |
| (**a**) | (**b**) |

**Figure 6.** (**a**) is a statistical line graph of user decision-making preferences in three modes, (**b**) is the distribution of extremely irrational decisions among different groups.

Then, we imported the data into a Statistical Package for the Social Sciences (SPSS) for significant analysis. First, the overall sample was tested for normality. As part of the sample data did not conform to the normal distribution, we chose to use non-parametric tests to test the data in the above figure from a horizontal and vertical perspective.

### 4.4.1. Transverse Analysis

Horizontally, we divided the data into four groups, including in three different modes of date and total data for analysis. The analysis method used primacy, middle and recency as samples and the Friedman test of k related samples. The test results are shown in Figure 7 below. The decision bias caused by the serial position effect in the overall decision-making process had marginal significance, proving that the serial position effect can induce human–computer interaction decision bias. Among them, the subjects in Mode one were affected by the recency effect with obvious significance (Mode one $p = 0.003 < 0.05$), the subjects in Mode two were affected by the recency effect and showed marginal significance (Mode two $p = 0.051 < 0.1$), and the subjects under Mode three were affected by the primacy effect and showed marginal significance (Mode three $p = 0.081 < 0.1$).

### 4.4.2. Vertical Analysis

In the longitudinal direction, we divided the data into three parts according to "primacy", "middle" and "recency" for testing, respectively. The analysis method used three patterns as samples, and we used the Kruskal–Wallis test of k independent samples. The test results are shown in Figure 8. The subjects' preference for primacy in the three modes was significant ($p = 0.001 < 0.05$), the middle was not significant ($p = 0.285 > 0.1$) and recency was significant ($p = 0.035 < 0.05$). The longitudinal data test results show that the decision bias induced by the serial position effect was affected by different evaluation modes, and primacy was the most affected, followed by recency. In order to compare the impact of different variables on user decision making in more detail, we used the Mann–Whitney test of two independent samples to compare the data results of the three modes in pairs. Comparison results for Mode one vs. Mode three, and Mode two vs. Mode three show significant differences in primacy and recency (Mode one vs. Mode three: primacy $p = 0.001 < 0.05$; recency $p = 0.025 < 0.05$. Mode two vs. Mode three: primacy $p = 0.006 < 0.05$; recency $p = 0.025 < 0.05$), none in the middle (Mode one vs. Mode three: middle $p = 0.217 > 0.1$.

Mode two vs. Mode three: middle $p = 0.141 > 0.1$) and no significant differences in the three positions of Mode one and Mode two (Mode one vs. Mode two: primacy $p = 0.394 > 0.1$; middle $p = 0.773 > 0.1$; recency $p = 0.954 > 0.1$).

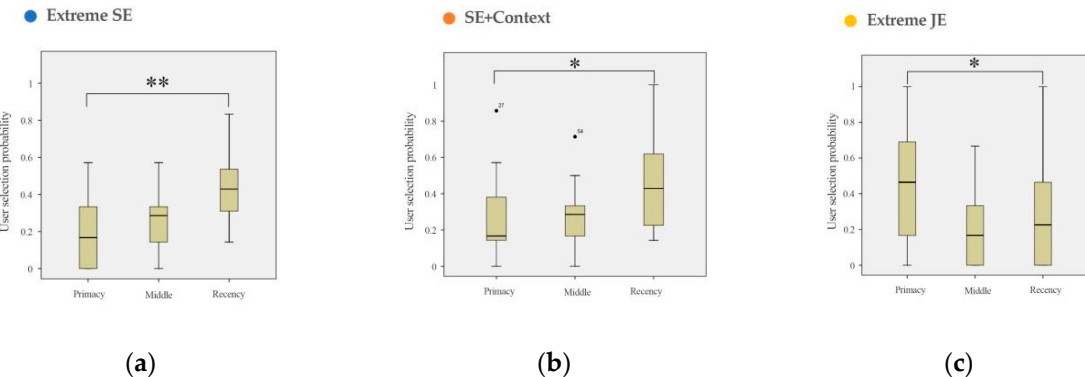

(**a**)　　　　　　　　　　　　(**b**)　　　　　　　　　　　　(**c**)

**Figure 7.** Significance test chart of three horizontal modes (** represents $p < 0.05$, * represents $p < 0.1$). (**a**) is the probability distribution diagram of three positions being selected in Mode 1, (**b**) is the probability distribution diagram of three positions being selected in Mode 2 and (**c**) is the probability distribution diagram of three positions being selected in Mode 3.

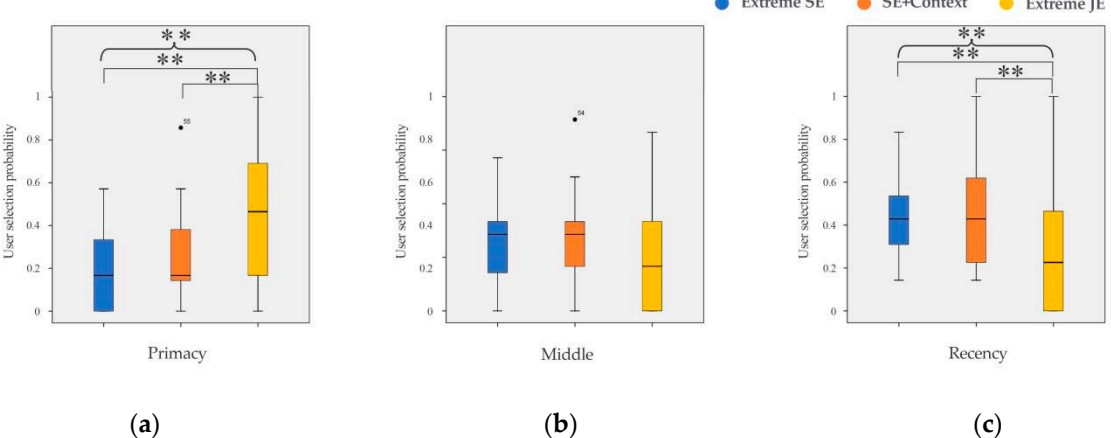

(**a**)　　　　　　　　　　　　(**b**)　　　　　　　　　　　　(**c**)

**Figure 8.** Longitudinal data on Kruskal–Wallis test results (** represents $p < 0.05$; braces contain three modes, and brackets contain two modes). (**a**) is the comparison of data in the three modes, which selected primary; (**b**) is the comparison of data in the three modes, which selected middle; and (**c**) is the comparison of data in the three modes, which selected recency.

### 4.5. Discussion

(1)　In this experimental environment, the serial position effect induced human–computer interaction decision-making bias, and the decision-making preference situation formed after the impact conformed to the serial position curve;

(2)　The user's decision evaluation environment affected the human–computer interaction decision bias induced by the serial position effect. The decision bias in SE was more affected by the recency effect, and when the user was under JE, they were more susceptible to the primacy effect;

(3)　Extreme SE showed a much higher probability of extremely irrational decisions than Extreme JE, and the context could effectively alleviate the errors generated under SE;

(4)　Among the two factors that make up the view, the information presentation significantly affected the decision bias of human–computer interaction induced by the serial position effect. In contrast, the quantity of information changed by the context had no significant impact on this decision bias.

## 5. Experiment Two

### 5.1. Objective

The primary objective of this experiment was to explore the impact of changing the context by folding detailed information on the human–computer interaction decision-making bias induced by the serial position effect. The mouse hovering method was added to compensate for the failure to collect clear eye movement paths in the previous groups of experiments and explore the difference between the sequence of user reception and the sequence of information presentation.

### 5.2. Experiment Setting

Experiment two supplemented the design of Mode four. On the basis of Mode three, the interactive operation of mouse hovering was introduced to complete the folding of detailed information. The experiment group was selected as the blank group reserved in Experiment one (this is compared with the grouping and presentation of Mode three, as shown in Table 2 and Figure 9).

**Table 2.** Experiment 2, specific grouping table.

| Experimental Group (7 Participants Each) | | Simulated Serial Number | | | | | | | | Information Independence | Quantity of Information |
|---|---|---|---|---|---|---|---|---|---|---|---|
| Group A | Mode 4 | 15 | 16 | 17 | 18 | 19 | 20 | 21 | 22 | Collapse | 1 + (5) |
| | Mode 3 | 23 | 24 | 25 | 26 | 27 | 28 | 29 | 30 | Card | 6 |
| Group B | Mode 4 | 23 | 24 | 25 | 26 | 27 | 28 | 29 | 30 | Collapse | 1 + (5) |
| | Mode 3 | 15 | 16 | 17 | 18 | 19 | 20 | 21 | 22 | Card | 6 |
| Group C | Mode 4 | 1 | 2 | 3 | 4 | 5 | 6 | 7 | | Collapse | 1 + (5) |
| | Mode 3 | 8 | 9 | 10 | 11 | 12 | 13 | 14 | 15 | Card | 6 |
| Group D | Mode 4 | 8 | 9 | 10 | 11 | 12 | 13 | 14 | 15 | Collapse | 1 + (5) |
| | Mode 3 | 1 | 2 | 3 | 4 | 5 | 6 | 7 | | Card | 6 |

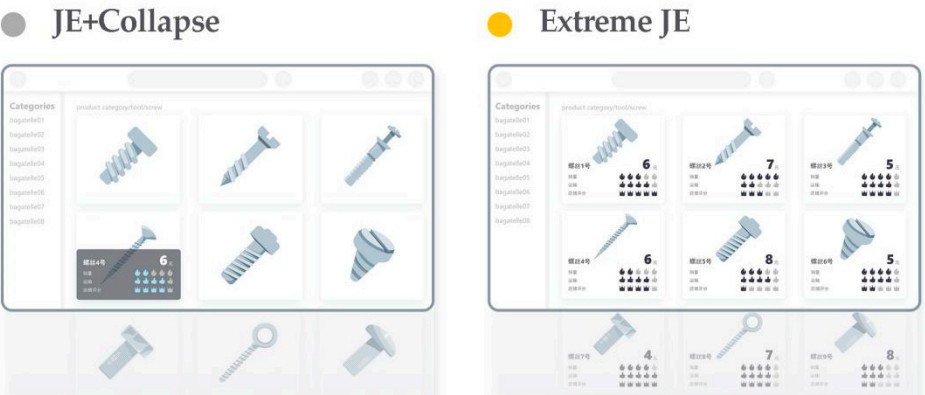

**Figure 9.** Comparison of Mode 3 and Mode 4 in a single product category.

### 5.3. Experiment Procedure

The experiment task was basically the same as Mode three in Experiment two, and there was an interactive operation of hovering the mouse to expand the detailed information of the target.

### 5.4. Analysis and Results

#### 5.4.1. Behavioral Data

The experiment recovered 23 groups of effective behavior data, with a total of 155 clear and compelling pieces of data. Compared with Mode three, Mode four had more on-screen feedback on mouse hovering; therefore, it was more convenient for data collection and processing. The processing result is shown in Figure 10. Nearly half of the subjects' browsing habits were in a "Z" shape (as shown in Figure 10a); that is, the sequence presentation order was equal to the subject's sequence reception order. However, more than one-third of the subjects' browsing habits were viewed in a horizontal "U" shape (as shown in Figure 10b). In addition to the two most conventional regular browsing methods, the browsing method with the highest repeat ratio ranking combined the first two methods. Through the observation of the subjects during the experiment, we can see that the subjects' browsing habits were more inclined to "Z"-shaped browsing when the page was fixed. Moreover, when the information needed to be viewed by sliding the mouse wheel, users were more inclined to change from "Z"-shaped browsing to horizontal "U"-shaped browsing (as shown in Figure 10c).

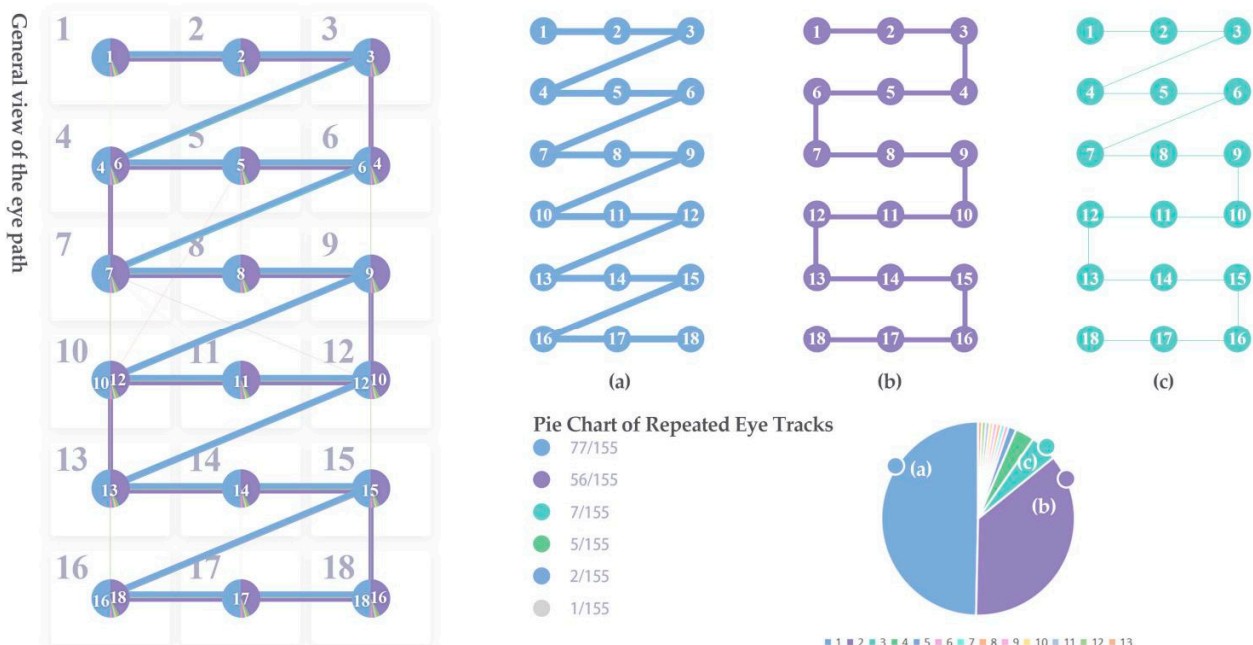

**Figure 10.** Eye tracking experiment data graphs, (**a**–**c**) are the three most common eye movement paths in order.

#### 5.4.2. Decision-Making Data

A total of 28 groups of valid subjective evaluation data were collected in the experiment. Excluding the results of one exercise for each group, which were not included in the statistics, a total of 192 valid decision data were collected, eight of which were extremely irrational decisions, and the probability of error was between Mode two and Mode three. This section compares the remaining 184 valid data with the overall trend of Mode three as follows. As shown in Figure 11, although Mode four was presented in the same way as Mode three, it was different from the primacy effect presented by Group three due to the different amount of information and interaction methods. Instead, there was a more obvious recency effect.

#### Transverse Analysis

The horizontal analysis method was the same as Experiment one, using the Friedman test of k related samples. The test result is shown in Figure 12. The recency effect

presented by the horizontal trend presented in Mode four was not significantly significant ($p = 0.391 > 0.1$).

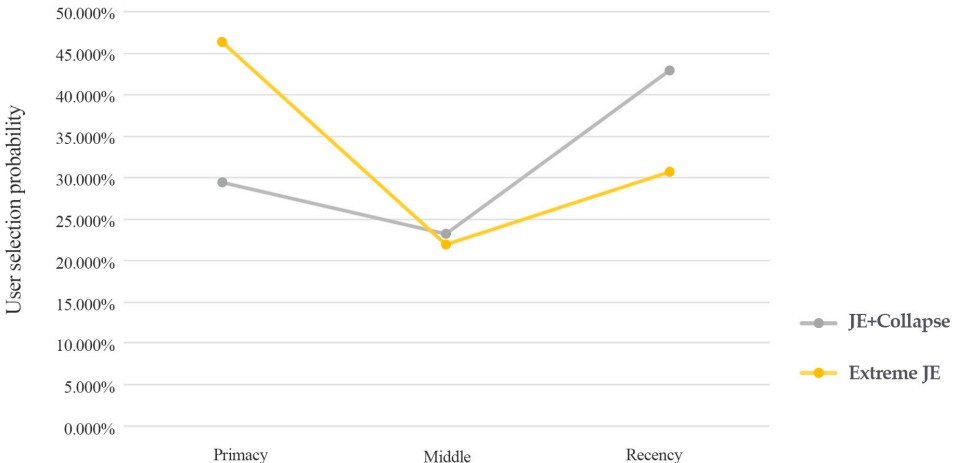

**Figure 11.** Comparison of experimental results between Mode 3 and Mode 4.

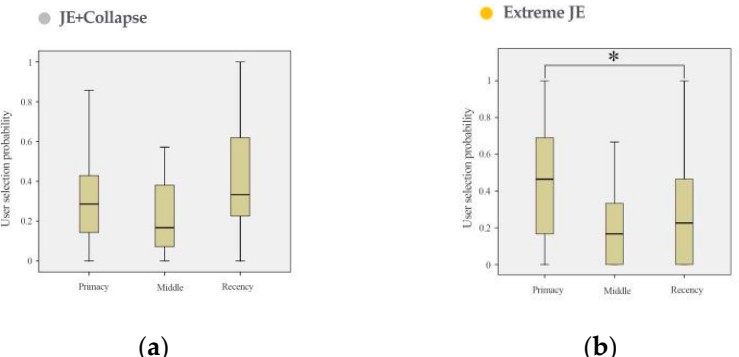

**Figure 12.** Mode 3 and Mode 4, horizontal analysis. (* represents $p < 0.1$). (**a**) is the probability distribution diagram of three positions being selected in Mode 4, (**b**) is the probability distribution diagram of three positions being selected in Mode 3.

Vertical Analysis

In the longitudinal direction, we divided the data of Mode three and Mode four into three parts according to "primacy", "middle" and "recency", and performed the Mann–Whitney test of two independent samples. The test results are shown in Figure 13. The difference was significant for primacy ($p = 0.026 < 0.05$), it was not significant in the middle ($p = 0.708 > 0.1$), and it was marginally significant for recency ($p = 0.054 < 0.1$).

*5.5. Discussion*

(1)　The experiment proves that the user's receiving sequence order was not exactly equal to the information presentation order, and the experimental results presented three regular user browsing paths;

(2)　After changing to collapse, the user's decision result was more affected by the primacy effect than the recency effect;

(3)　Under the waterfall flow layout, changes in explicit and implicit details significantly impacted the user's decision-making preferences in different locations.

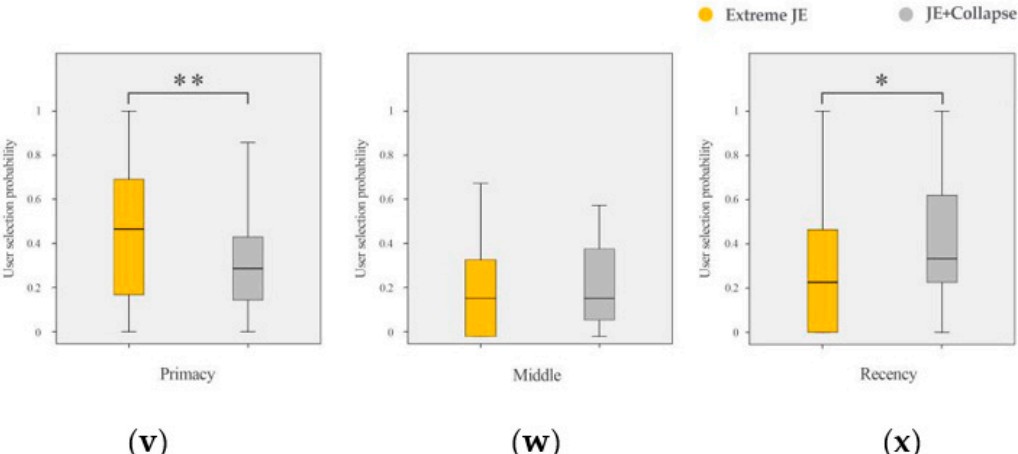

**Figure 13.** Longitudinal data on Mann–Whitney test results.

## 6. Discussion

### 6.1. User's Sequence Receiving Order

Except for "Z" browsing, which accounted for 49.7% of total behavior, other browsing habits will disrupt the sequence presented by the designer and cause the uncontrollability of the user's receiving sequence. This experiment was designed to avoid the influence of these conventional browsing methods on the experimental results (the distance between the experimentally set optimal values was greater than or equal to three, and the number of stimuli presented in each row was three. Therefore, the three optimal values we set under several conventional browsing habits, "primacy", "middle" and "recency", did not affect the order of presentation; therefore, the decision data analysis remains unchanged. However, the user's different browsing habits will result in a different received sequence or order in actual applications. The user's regular eye movement path extracted in this experiment is helpful to understand and explain the difference between the sequence presentation order and the subject's sequence reception order in a more complex environment.

### 6.2. Decision Bias

According to the horizontal analysis in the two experiments, we can observe that the serial position effect affects the user's decision-making results, and the impact is mainly manifested in the user's ignorance of the intermediate position information. The horizontal analysis between the three mode groups in Experiment one confirmed that different views could cause a decision preference reversal between the primacy effect and the recency effect in the serial position effect. In addition, the user decision result under SE is more affected by the recency effect, and the user decision result under JE is more affected by the primacy effect. The horizontal analysis of Mode four in Experiment two shows that the change of explicit and implicit details will reverse the user's decision-making preference. Therefore, the previous hypotheses are all valid.

According to the longitudinal analysis in Experiment one, we can compare the two modes to explore how the two factors, information presentation and quantity of information, play a role in the reversal of preferences. According to the longitudinal analysis in Experiment two, a comparison is made between Mode three and Mode four to explore how the explicitness and implicitness of contextual information play a role in preference reversal in different positions. The analysis of the comparison results of the four modes is as follows:

**Mode one vs. Mode three:** Under the same experimental stimulus, the impact of the two different extreme evaluation modes (information presentation and quantity of information are not equal) on the user's decision-making results will have highly significant differences in primacy and recency. The effect on primacy preference in SE is significantly lower than in JE, while recency presents the opposite result;

**Mode one vs. Mode two:** With information presentation, changing the quantity of information by adding context alternatives does not have a significant impact on the final decision result of the user;

**Mode two vs. Mode three:** When the quantity of information is entirely equal, only changing the information presentation mode still affects the user's decision-making. There is a very significant difference between primacy and recency;

**Mode three vs. Mode four:** In the case of the same information presentation and the quantity of information, the detailed information in the context is hidden, and only the selected information is displayed by hovering the mouse. Compared with the full display of the information, the preference for the top-end will be significantly weaker.

From the longitudinal analysis results, we can conclude that the changes of different factors and conditions in the presentation of commodity information constitute a different evaluation mode for users, which affects the final decision-making preference reversal. By carefully studying the adjustment of factor levels in the experiment and the transformation of the evaluation mode, we can find the rules between different factors. When the evaluation mode in the overall environment is more separate, and the contextual influence is more negligible (carousel, the quantity of information is reduced, or collapse), the user's decision making tends to be affected by the recency effect. When the evaluation mode in the overall environment is more combined and contextual influence is more significant (card, the quantity of information is more, or without collapse), user decisions tend to be affected by the primacy effect. Combined with theoretical knowledge, in a broad sense, the environment constituted by the context effect affects the decision-making process in which the decision maker participates [46]. The contextual influence can be adjusted by changing how the product is presented, thereby adjusting the user's decision-making preferences.

*6.3. Application and Limitation*

The hypothesis that the serial position effect can induce user decision-making bias in information visualization has been confirmed in this study. In the research field, this can help researchers to analyze and evaluate the causes of user decision-making results in marketing management, and the research results can be applied to the design and development of shopping websites, which is conducive to online suppliers using this preference to induce the user's decision-making results. For example, more profitable products should be arranged in the advantageous position under the information presentation method and in the proper product presentation position. In addition, although our experimental research background selected the online shopping environment, the experimental settings are not affected by the particularity of the shopping environment and can be mapped to other complex information system interactions. In modern industrial production and control, aerospace, military command and combat, disease control and medical applications, the accuracy of decision-making judgments is a direct factor that affects the work of the entire system. Researchers can try corresponding correction methods for the law of context effect based on this research to alleviate user decision-making bias and induce users to make more rational decisions. In the following steps, we will further study the influence of form on the decision preference induced by the serial position effect and conduct a more in-depth exploration of the decision bias induced by the serial position effect.

**7. Conclusions**

In response to the three questions raised in the Introduction, this research obtained the following findings:

Finding one: This study proved that the sequence position effect can induce the human–computer interaction decision bias, and the user's decision result obviously fits the U-shaped serial position curve of the serial position effect. The reverse analysis of Experiment one supports this conclusion.

Finding two: The change of views in information visualization affect the decision-making bias in human–computer interaction induced by the serial position effect. The

three presentation methods selected in the experiment showed obvious and significant differences between the decision-making results. The vertical analysis of Experiment one supports this conclusion.

Finding three: This research showed that among the factors constituting different views of information visualization, information presentation, quantity of information and explicit and implicit details will all affect user decision-making results to varying degrees. We combined the conceptual analysis of the context effect on the decision-making environment. When the evaluation mode in the overall environment is more separate and the contextual influence is more negligible, the user's decision making tends to be affected by the recency effect. When the evaluation mode in the overall environment is more combined and contextual influence is more significant, user decisions tend to be affected by the primacy effect. The study also provided eye movement evidence, showing the difference between the user's receiving sequence and the interface presentation sequence, and showed several common eye movement paths for subsequent research reference.

Compared with previous studies, this article organically combines the "sequence position effect" and "human–computer interaction decision-making" for the first time, confirming the influence of this memory effect on decision making. Since the existing adjustment methods commonly used in the field of psychology are difficult to apply to human–computer interaction scenarios, such as, reducing decision-making bias by extending feedback time, shortening the length of information, increasing auditory channels, etc., we tried to find an effective way to adjust this decision-making bias from a more universal perspective (interface presentation). In the future, designers can choose appropriate presentation methods according to actual application requirements. Whether to use the user's decision-making preferences to maximize the benefits or avoid adjusting the decision-making bias to get closer to the ideal optimal decision can all be applied according to the needs of different scenarios.

This paper currently only studied the adjustment effect of the information presentation mode in the visualization of human–computer interaction information on the sequence position effect. In fact, information visualization is a large research field, and many levels of it will affect the decision-making results of users to varying degrees. In the next step, the team will take the form of information representation as the research point to study the moderating effect of the choice of different visualization forms on the decision-making bias induced by the sequence position effect.

**Author Contributions:** Conceptualization, Z.Z. and N.P.; methodology, Z.Z., N.P. and C.X.; software, Z.Z.; validation, Y.N., H.W. and C.X.; formal analysis, Z.Z. and N.P.; investigation, Z.Z. and N.P.; resources, C.X.; data curation, C.X.; writing—original draft preparation, Z.Z.; writing—review and editing, Z.Z. and N.P.; visualization, Z.Z.; supervision, Y.N., H.W. and C.X. All authors have read and agreed to the published version of the manuscript.

**Funding:** The paper is supported jointly by Natural Science Foundation of China (No. 71871056, 71801037 & 72171044) and Aerospace Science Foundation of China (No. 20200058069002).

**Institutional Review Board Statement:** Not applicable.

**Informed Consent Statement:** Informed consent was obtained from all subjects involved in the study.

**Data Availability Statement:** The data are contained within the article.

**Conflicts of Interest:** The authors declare no conflict of interest.

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
