# Peer review of "The Influence of Commodity Presentation Mode on Online Shopping Decision Preference Induced by the Serial Position Effect"

_applsci, doi:10.3390/app11209671_

Round 1

Reviewer 1 Report

The paper examines different ways of presenting a set of items, such as in online shopping, to understand how different presentations can influence user choice. The paper describes two experiments that were conducted on a sample of about 30 people. The first experiment aims to show that separate evaluation (of proposed items) produces decision biases and highlights the differences with joint evaluation. The second experiment aims to identify the implicit and explicit details that led to decision bias in the case of the serial position effect. Here, eye tracking is used, integrated with mouse hovering.

The structure of the paper leads into confusion, the design of the two experiments is in the state of the art, while it should be in a separate section, or together with the next section which describies the methodology. Section 4 describes the first experiment, while section 5 describes the second one. In section 5, transverse analysis and vertical analysis should be indicated as subsections, as in section 4.

In the description of the experiments the terms mode and model are used indifferently. They are two different words and should always be used the same (e.g. mode).

The experiments are quite articulated, a detailed analysis has been made between the different groups, and for each specific considerations have been made about the observed results. However, the final conclusions remain rather general and does not give a clear answer to the three Research Questions written in the introduction. The conclusions should be rewritten in a more organic way, underlining the innovative contribution that this work brings compared to the state of the art.

Summarizing: the work is difficult to read, in some points is not well structured. The experiment is detailed and articulated, but it is not clear how the results obtained bring a contribution to the scientific literature in the field. In other words: what is the added value of this contribution compared to the state of the art?

MINORS

In the sentence toward the end of page 3 "Our study first investigates ... and discussion." The meaning is unclear.

  1. 3, third last line of subsection 2.1: the acronym UI should be made explicit, please write User Interface (UI), for this first occurrence.

3.2.3 and 4.4.1 are all written in bold.

The caption in Figure 13 is highlighted in yellow.

First line of page 11:

"...we imported the data into SPSS..."  ->"...we imported the data into Statistical Package for the Social Sciences (SPSS)...". Possibly you should also add a reference or footnote.

Second line of section 4.1.1

"...data and data..." -> "... data...". If it is not an error you should rewrite the sentence, because it is not clear.

Reviewer 2 Report

Thank you for the opportunity to review this interesting article. After reading the entire article I found the following issues that should be clarified related to:

1. Abstract. The authors do not clearly state from the beginning the main purpose of their research, but only how they will achieve this. I suggest the authors do this!

2. Introduction. It is brief and the authors provide additional clarification on the main objectives of their research, including research questions to which they wish to provide supporting answers.

3. Related Work. This section should be simply renamed "Literature review". The 2 subsections are presented quite clearly but there are also other studies of specialists who have contributed in this regard and who I think should be added in this section such as:

Oncioiu, I., Căpușneanu, S., Topor, DI, Tamaș, AS, Solomon, A.-G., Dănescu, T., Fundamental Power of Social Media Interactions for Building a Brand and Customer Relations, Journal of Theoretical and Applied Electronic Commerce Research, 2021, 16 (5), 1702-1717, https://doi.org/10.3390/jtaer16050096

Liao, C., To, P., Wong, Y., Palvia, P.C., & Kakhki, M.D. (2016). The Impact of Presentation Mode and Product Type on Online Impulse Buying Decisions. Journal of Electronic Commerce Research, 17, 153.

Koch, J .; Frommeyer, B .; Schewe, G. Online Shopping Motives during the COVID-19 Pandemic — Lessons from the Crisis. Sustainability 2020, 12, 10247. https://doi.org/10.3390/su122410247

Di Crosta A, Ceccato I, Marchetti D, La Malva P, Maiella R, Cannito L, et al. (2021) Psychological factors and consumer behavior during the COVID-19 pandemic. PLoS ONE 16 (8): e0256095. https://doi.org/10.1371/journal.pone.0256095

4. General methodology. This section should be simply renamed "Research methodology". The authors specify that they used the questionnaire for research but do not present the questions that were addressed to the subjects. I suggest the authors to present in an appendix the questions contained in the questionnaire designed by them! The number of participants in this questionnaire is relatively small, 30 participants!

Figure 6 (a) is not clear enough to understand the details! I suggest the authors to do this as well as those mentioned above!

5. General discussions. This section should be simply renamed "Discussions".

6. Conclusions. In this section the authors do not present the limits of their research nor the future directions proposed by the research. I suggest the authors do this!

Round 2

Reviewer 2 Report

I am very pleased with the authors respinse mailing my all suggestions.